# Experience with Obese Patients Followed via Telemedicine in a Latin American Tertiary Care Medical Center

**DOI:** 10.3390/ijerph191912406

**Published:** 2022-09-29

**Authors:** Alejandro López, Maria Fernanda Escobar, Alejandra Urbano, Juliana Alarcón, Laura Libreros-Peña, Diana Marcela Martinez-Ruiz, Luz Ángela Casas

**Affiliations:** 1Department of Endocrinology, Fundación Valle del Lili, Cali 760032, Colombia; 2Department of Telemedicine, Fundación Valle del Lili, Cali 760032, Colombia; 3Department of Gynecology and Obstetrics, Universidad Icesi, Cali 760031, Colombia; 4Centro de Investigaciones Clínicas, Fundación Valle del Lili, Cali 760032, Colombia

**Keywords:** obesity, telemedicine, eHealth, overweight, health care access, telehealth

## Abstract

Background: Obesity is a major public health concern worldwide. Latin America has experienced rapid growth in obesity incidence during the last few decades. Driven by confinement measures, a telemedicine program was implemented in March 2020 to give continuity to obese patients’ care through a weight loss program led by the endocrinology department in a tertiary care medical center in Latin America. Objective: This study aimed to describe the clinical experience of using digital health for monitoring and attention of obese patients and description of weight change outcomes of these patients followed via telemedicine during March 2020–December 2020. Methods: A retrospective cohort study was conducted including 202 patients. A Skillings-Mack test was performed to conduct a subgroup analysis of the medians of the weight over the follow-up period, and a mixed multiple linear regression model was performed to estimate the expected average change in weight over time Results: We observed good adherence to the program, represented by a weight loss of −4.1 kg at three months of follow-up, which was maintained even during the sixth month of follow-up. Conclusions: Digital Health strategies such as telemedicine can be a helpful tool for both patients and health care providers to support the continuity of care and showing satisfactory results in the management of obese patients.

## 1. Introduction

Obesity is a serious complex chronic disease and a major public health problem; approximately 39% of the world’s population is overweight or obese, representing an increase in the worldwide prevalence of nearly 50% in the last 35 years [1]. In Colombia, 56.5% of adults are overweight, and 18.7% meet the obesity criteria [2]. This disease is linked to multiple multisystemic complications, such as type 2 diabetes, cardiovascular diseases, malignancies [3], and an increased risk of dying from any cause [4], thus posing a significant socioeconomic burden and necessitating the development of strategies to manage obesity and overweight [5].

The novel coronavirus pandemic and the measures adopted to reduce the spread of the virus have led to several lifestyle changes affecting dietary habits, physical health, mental health, and body weight [6]. Individuals with preexisting concerns such as obesity, eating disorders, and mental health conditions are particularly vulnerable to lockdown effects [7]; reduced contact with their health care providers during the lockdown led to a decline in the management of their disease and treatment adherence [8].

Telemedicine is defined as the use of electronic information and communication technologies to provide and support health care [9]; it has been recognized as an invaluable tool for reducing barriers to access and allowing long-term follow-up of patients even before the COVID-19 pandemic and has led to an increase in patient satisfaction, accessibility, and financial savings for patients and health care systems [10,11]. With the development of technology, communication strategies, and the internet, healthcare providers can implement different telemedicine approaches, making telemedicine a rapidly growing field that has expanded in recent years [12].

Strategies to control and monitor overweight and obese patients using telehealth have shown good results, allowing providers to guide and promote healthy lifestyles and provide behavioral recommendations through this care modality using email, text messages, websites, or phone calls, leading to patient weight losses of up to 5–10% [13,14,15], with even more satisfactory results when self-monitoring strategies are added [16,17]. These small weight changes (5–10%) lead to a decrease in metabolic complications associated with the disease [18,19]. However, evidence and interventions based on synchronous telemedicine tools or involving real-time video conferencing have not been conducted in populations other than Latin Americans.

“Siempre” is an outpatient teleconsultation care model implemented in Fundación Valle del Lili (FVL), a tertiary Latin American university hospital in March 2020 driven by the lockdown measures established in Colombia to provide continuity and care to obese patients during this time. We aim to describe the effect of this consultation modality on the weight and other metabolic parameters of obese patientswith measurements of the first 3 months of implementation and 6 months later.

## 2. Materials and Methods

### 2.1. Design and Context

We conducted a retrospective cohort study of patients with obesity managed through the teleconsultation program from 1 March to 31 December 2020. FVL is a tertiary care academic medical center located in the southwestern region of Colombia.

In response to the national mandatory restrictions imposed due the COVID 19 pandemic, FVL implemented an ambulatory teleconsultation program, conducted via videoconference tools in real-time, the endocrinologists evaluated all patients, and the treatment and time between teleconsultations were individualized.

### 2.2. Overview of the “Siempre” Teleconsultation Program

The patient communicates through a designated institutional telephone number, and an agent schedules the appointment with an endocrinologist on the Microsoft Teams^®^ platform. The day of the appointment, the patient is contacted 30 min before to verify their identity, capture a photo of the patient for identification purposes, fill out an informed consent form, and authorize the handling of personal data by the institution. Medical attention through this tool addressed by the specialist is focused on nutritional recommendations, monitoring of comorbidities such as Diabetes Mellitus, Hypertension, and dyslipidemia; revision and request for paraclinical tests, and guidance in lifestyle change. The data from the teleconsultation are registered in the institutional clinical record system (SAP). After the consultation, a summary of the appointment is sent by email in pdf format along with the medical prescription.

The paraclinical tests ordered by the doctor for the physician include a lipid profile, uric acid, ferritin, blood glucose levels, and testosterone for male patients, also including an order of dual-energy X-ray absorptiometry (DEXA), to be evaluated during the controls.

Testosterone is a key hormone in metabolic diseases such as obesity, where decreased levels are associated with increased corporal mass; it has also been related to an energy imbalance with metabolic effects such as decreased insulin sensitivity and dyslipidemia among male patients [20]. Ferritin was ordered to evaluate iron deposits and elucidate a possible established non-alcoholic fatty liver disease (NAFLD) [21].

This study was conducted according to the Declaration of Helsinki guidelines; the institutional review board reviewed and approved the study protocol (IRB/EC No. 1749; Act No. 07-2021). Informed consent was not required due to the retrospective nature of the study, which was classified as a risk-free study according to resolution No. 008430 of 1993, article 11, numeral A of the Ministry of Health and Social Protection of Colombia.

### 2.3. Population and Sample Size

We included adult patients (>18 years old) who had been diagnosed with and/or treated for obesity and who had received at least two outpatient follow-ups with endocrinology specialists from March 2020 to December 2020. We excluded patients who were attended only once by teleconsultation, patients who had two teleconsultations in the same month without clear specifications, and patients who required face-to-face medical attention due to their clinical condition or if vital signs and a physical examination were crucial for decision-making.

### 2.4. Variables

All information was collected retrospectively from institutional medical records and registered in the software program Clinic (Fundación Valle del Lili, Cali, Colombia). We collected sociodemographic and clinical data, including sex, age, health security, job occupation, and comorbidities. Weight and other metabolic parameters were evaluated three times: at the first teleconsultation (time 0), at three or fewer months (time 1), and at six or fewer months (time 2). Eating habits and physical activity were subjectively considered by the clinicians and was also obtained from medical records.

### 2.5. Statistical Analysis

We summarized the qualitative variables with absolute frequencies and percentages; the quantitative variables were presented with the measures of central tendency (mean or medians) and their respective measures of dispersion (standard deviation or interquartile ranges) according to the obtained distribution with a Shapiro–Wilk test. A subgroup analysis was performed using a Skillings-Mack test to determine changes in weight during the different time periods, subgroups were determined by age, type (first-time consultation or follow-up), number of teleconsultations, and stage according to the diagnostic criteria for adiposity-based chronic disease (ABCD).

The Skillings Mack test is an extension of Friedman’s test that allows for missing values; this was chosen because some patients did not have all measurements. To estimate the average change in weight over time, we performed a mixed linear regression model with random intercept used to control inter-patient weight variability. Variable selection for multivariate models was performed using the “purposeful variable selection” approach described by Bursac et al. [22]. Initially this method fits the univariate model with each variable considering an initial significance (*p* < 0.25), variables are kept in the model if they tend to confound the iterative process.

The data analysis was performed using RStudio Team (2020). RStudio: Integrated Development for R. RStudio, PBC, Boston, MA, USA, URL http://www.rstudio.com/, we performed a STATA^®^ data analysis for descriptive statistics.

## 3. Results

A total of 510 subjects were diagnosed with obesity during the study period. We excluded 308 subjects, of whom 294 registered for one teleconsultation, 12 had two teleconsultations in the same month, and 2 were minors. Finally, 202 patients were included in the final analysis; 88 (45.6%) of them were receiving a consultation for the first time, and 114 (54.4%) had been evaluated previously by in-person consultation (Figure 1).

The median age of the first-time group was 39 (IQR 32–45) years, and for patients who required medical follow-up, it was 41 (IQR 33–52) years. A total of 69.8% (141) of the total sample (65.9% (58) of the first-time group [FG] and 72.8% (83) of the follow-up group [FUG]) had comorbidities, with thyroid diseases (14.15%), arterial hypertension (13.86%) and dyslipidemia (12.37%) being the most prevalent illnesses. At the first teleconsultation, 56.8% (50) in FG and 31.6% in FUG (36%) did not perform physical activities, and 72.7% (64) in FG and 41.2% (47) in FUGG did not have a healthy diet. Table 1 summarizes the remaining sociodemographic and clinical characteristics.

At the first evaluation, patients in the first-time consultation group registered a median weight of 82 (IQR 73–92) kg and a body mass index of 30.3 (IQR 27.5–33.7) kg/m^2^, and patients in the follow-up consultation group had a median weight of 80.2 kg (IQR 71.2–97) and a BMI of 30.4 (IQR 26.55–33.5) kg/m^2^. In terms of the ABCD stage, most patients were classified as stage 1, and stage 2 was only identified in the FUG of 5.3% (6) of the subjects. Table 2 shows the values of the other metabolic parameters at the first encounter, and Table 3 recompiles values of the same measurements at time 1 (3 or fewer months) and time 2 (6 or fewer months).

Last, we included 78 patients to identify the change in weight at the follow-up; 63 patients had their last control at time 1, and 36 patients had it at time 2. Figure 2 shows the decrease in weight over time, with a median at time 0 of 84 kg (IQR: 73–97), at time 1 of 78 kg (IQ: 68–90), and at time 2 of 75 kg (6 IQR: 68–88). Additionally, a subgroup analysis was performed to assess the change in weight over time, which allowed us to find statistically significant weight loss in relation to age, type of teleconsultation, the number of visits, and ABCD stage of the disease, except in patients older than 60 years and in ABCD stage 2 (Table 4).

Table 5 shows the mixed multiple linear regression model. At time 1, there was a reduction of 4.1 kg (95% CI: −5.1, −3.1; *p* < 0.001), and at time 2, a reduction of 8.6 kg (95% CI: −10, −7.3; *p* < 0.001) was observed. In addition, the ABCD stage is a factor that increases body weight; namely, at a higher stage, we expected an increase in the average weight.

## 4. Discussion

### 4.1. Principal Findings

Patients diagnosed with obesity who continued to control their disease during the first months of the institutional outpatient teleconsultation program registered statistically significant weight loss, with no confounding effect on age, number of visits, type of teleconsultation (first-time evaluation or control patient) or ABCD stage, which was maintained at the 6-month evaluation. Patients older than 60 years and those classified as ABCD stage 2 were the only categories that did not reach statistical significance.

We observed a weight loss of −4.1 kg at three months of follow-up, which was statistically significant concerning the weight at admission.

### 4.2. Results in Context

Latin America has some of the highest obesity rates in the world, where the transition from an underweight population to overweight and obese population has been relatively rapid [23]. The definition of obesity is changing, and previous models were based primarily on BMI. In recent years, the American Association of Clinical Endocrinologists (AACE) and the American College of Endocrinology (ACE) have proposed the term adipogenesis-based chronic disease (ABCD), which approaches obesity as a chronic disease, proposing 4 key elements for patient management: (1) lifestyle modifications, (2) standardizing protocols for durable weight loss and management of complications, (3) contextualization of patient care aiming to control obesogenic factors in the patient environment, and (4) development of strategies based on evidence for successful implementation, monitoring and optimization of patient care [24].

Previous studies conducted before and after the COVID-19 pandemic have focused on the effectiveness of digital health interventions, although telemedicine was already in use before the pandemic. Confinement measures and social distancing have led to a rapid expansion in the use of telemedicine, improving access to care, reducing patients’ waiting times, and making it more acceptable to both health care professionals and patients [11,25].

In 2017, Alencar et al. conducted a randomized clinical trial, where an accelerometer, a blood pressure monitor, and a scale were used to assess body composition. Additionally, patients in the intervention group had a monthly evaluation with an endocrinologist and weekly counseling with nutritionists, both carried out by videoconference. Twenty-five patients were randomized 1:1 in both groups and followed for three months. Again, the group that received advice via telemedicine registered a greater decrease in weight (intervention 7.3 kg [±4.4] vs. control 1.5 kg [±4.1]; *p* = <0.005). Additionally, a greater decrease in the percentage of body fat was recorded in the intervention group (−9 ± 8.3% vs. 1.3 ± 7.7%; *p* < 0.05) [26]. Although self-monitoring is an effective tool in weight loss, we consider that specialized follow-up by an endocrinologist through videoconferencing, such as the intervention performed in our investigation, can enhance these results, encouraging users and supporting greater adherence to recommendations.

Johnson et al. carried out a randomized clinical trial in which they compared a weight loss program delivered via VC with that delivered in person (IP). Ten participants were randomly assigned to each of the groups and followed for 12 weeks. The VC group achieved a more significant loss (CV −8.23 kg [±4.5] vs. PI −2.9 kg [±3.9]; *p* < 0.05), while no statistically significant differences were found in glycemia and glycosylated hemoglobin levels. The authors attribute these results mainly to the ease of adherence facilitated by telemedicine; in their case, 100% of the VC group attended the counseling sessions compared to 80% of the IP group [27]. One of the main differences from our study is that we did not have self-monitoring tools that providers could follow, which may be one reason why the patients in our study lost less weight than those in Johnson’s study. Nevertheless, our results evidence that counseling delivered via teleconsultation leads to significant weight loss, which was even more remarkable than that reported in the face-to-face program, which could have been influenced by the confinement measures since more teleconsultations were conducted in this period than face-to-face assistance. However, our results adhere to the fact that the emergence of digital health applications is a promising opportunity to close the current coverage gap for patients with obesity [28].

Recent evidence has shown that the use of digital media through different platforms affects behavioral changes, the most effective health behavior changes interventions combine both digital and in-person components [29]. The global health strategy 2020–2025 aims to strengthen digital health networks to promote access to health services and promote higher standards of health and access services [28]; taking this into account with the gradual recovery of life routines, the design of these strategies will allow for long-term follow-ups, approaching an accurate assessment of the effects of these interventions. The World Health Organization (WHO) is carrying on new strategies to promote digital health solutions to decrease obesity worldwide, since digital health provides valuable options to enhance well-being [30].

### 4.3. Limitations and Strengths

The retrospective nature of this study is a limitation. Additionally, other variables, such as the type of physical activity, individual nutritional changes, and weight loss medications, were not assessed individually, although these variables might affect the results; nevertheless, being a self-controlled study design represents another limitation of our study, due the lack of a control group. However, there are no other investigations that evaluate weight management focused on stage ABCD through telemedicine or that include the Latin American population, so it is not possible to compare our results. Still, we show that patients in stage 2, despite not achieving a significant reduction in weight, were the group that showed the greatest weight loss in kg, which adheres to the recommendations of societies for reducing complications and achieving greater metabolic control.

Nevertheless, this is the first study conducted with a Latin American population, aiming to understand the potential role of a digital health strategy such as telemedicine as a feasible tool for the care and follow-up of obese and overweight patients in a Low-middle income context.

### 4.4. Future Implications

Obese patients have shown a tendency to regain weight after the end of treatment, representing a major concern in obesity management [31,32]. During the intervention in this study, the number of patients at the end of follow-up with good adherence to the diet and physical activity recommendations increased for both groups. Healthy lifestyles and behavioral recommendations are the first lines of treatment for obese patients [33]. The development of follow-up tools, seeking to maintain the lost weight by establishing objectives and allowing measurements of patient progression, are also goals of the new proposals for obesity management [14,34]. Telemedicine and mobile health promote lifestyle modifications and healthy lifestyles, being relatively inexpensive tools, and improving access to health care for patients. While not intended to replace physical clinical attention, they offer additional tools to monitor and follow up with patients [8,35], ensuring that patients do not give up on their new habits in the long term.

In our study, patients older than 60 years did not achieve significant weight loss. Very few studies have focused on the weight loss of elderly patients, conferring a critical gap in knowledge [36]. Few studies that have been carried out focused on elderly patients have shown losses of up to 5% of the initial weight, improvement in lifestyles, and good satisfaction with the use of tools [37]. Therefore, further research focused on this population is necessary.

## 5. Conclusions

This study demonstrated that the use of technologies and care models such as telemedicine represents good support for patients and healthcare providers in obesity management, showing good adherence and weight outcomes derived from this intervention. Obesity will continue to be a public health problem in the coming years. Medical attention through telemedicine, mHealth, and virtual care can reduce barriers to accessing good strategies to maintain patient adherence and commitment to weight loss programs. Further studies which evaluate the cost-effectiveness of telemedicine interventions in Latin America could provide economical insight into the possible benefits of these interventions to public health care.

## Figures and Tables

**Figure 1 ijerph-19-12406-f001:**
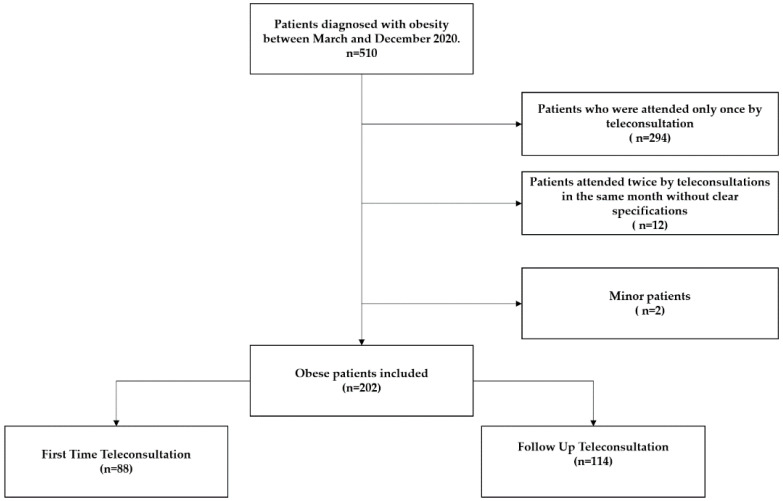
Included Patients Flow-Chart.

**Figure 2 ijerph-19-12406-f002:**
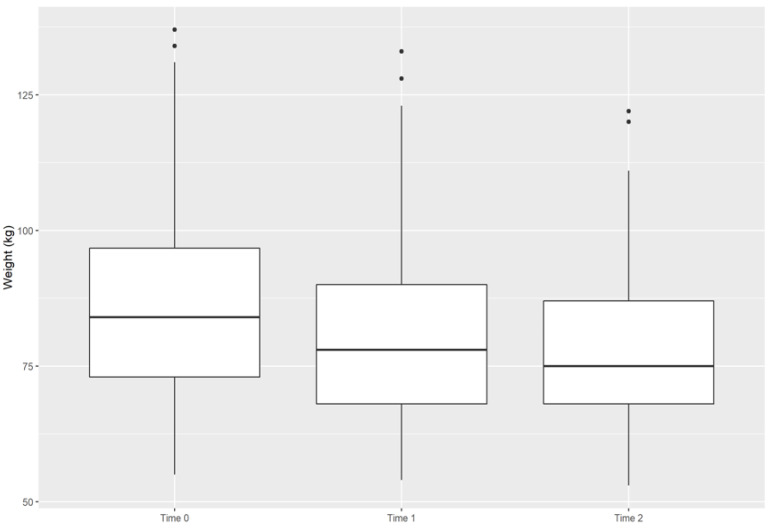
Box plot graphic showing the median and interquartile range of weight at each time point.

**Table 1 ijerph-19-12406-t001:** Sociodemographic and clinical characteristics.

	First-Time Group (FG), n = 88	Follow-Up Group (FUG), n = 114
Number of teleconsultations *	3 (2–5)	3 (2–4)
Age, years *	39 (32–45)	41 (33–52)
Health Insurance%)		
Contributive	11 (12.5)	11 (9.6)
Prepaid	71 (80.6)	101 (88.5)
Particular	6 (6.81)	2 (1.71)
Residence (%)		
Urban	80 (90.9)	111 (97.3)
Rural	5 (5.6)	-
ND	3 (3.4)	3 (2.6)
Occupation (%)		
Unemployed	3 (3.4)	4 (3.5)
Employee	55 (62.5)	67 (58.7)
Independent	6 (3.4)	7 (6.1)
Home	3 (3.4)	6 (5.2)
ND	21 (23.8)	30 (26.3)
School grade		
High school	1 (1.1)	2 (1.7)
College	4 (4.5)	9 (7.8)
Postgraduate	43 (48.8)	63 (55.2)
Analphabet	10 (11.3)	5 (4.3)
ND	30 (34.0)	35 (30.7)
Comorbidities (%)		
No	30 (34.0)	31 (27.1)
Hypertension	8 (9.0)	20 (17.5)
Diabetes	2 (2.2)	6 (5.2)
Dyslipidemia	10 (11.3)	15 (13.1)
Coronary artery disease	-	1 (0.8)
Thyroid disease	14 (15.9)	15 (13.1)
Fatty liver	4 (4.5)	10 (8.7)
Physical activity at first teleconsultation (%)		
No	50 (56.8)	36 (31.5)
Yes	23 (26.1)	57 (5)
ND	15 (17.0)	21 (18.4)
Healthy diet at first teleconsultation (%)		
No	64 (72.7)	47 (41.2)
Yes	7 (7.9)	49 (42.9)
SD	17 (19.3)	18 (15.7)

* Median (IQR).

**Table 2 ijerph-19-12406-t002:** Clinical characteristics at the first teleconsultation by type of evaluation.

	First-Time Group (FG) n = 88	Follow-Up Group (FUG) n = 114
Weight, kg *	82 (73–92)	80.2 (71.25–97)
BMI, kg/m^2^ *	30.3 (27.5–33.7)	30.4 (26.55–33.5)
ABCD stage (%)		
stage 0	43 (48.8)	36 (31.5)
stage 1	44 (50)	72 (63.1)
stage 2	1 (1.13)	6 (5.26)
Percentage of body fat by DEXA	44 (40.4–49)	46.6 (45–48.5)
Total fat weight by DEXA, kg	38 (32–42.5)	34.5 (31.1–37.95)
Android fat/gynoid fat ratio by DEXA	1.025 (0.8–1.1)	0.96 (0.9–1)
Fasting blood glucose (mg/dL)	91.6 (84–98)	91 (85.5–100)
LDL colesterol (mg/dL)	115.25 (97.3–131.5)	117 (97–139)
HDL colesterol (mg/dL)	48.9 (42–57.9)	46 (40–57)
Triglycerides (mg/dL)	134 (90–198)	121 (88–166)
Creatinine (mg/dL)	0.76 (0.6–0.8)	0.78 (0.68–0.8)
Ferritin (ng/mL)	92.5 (45–185.5)	94.9 (44.9–222)
Glycosylated hemoglobin (%)	5.4 (5.2–5.6)	5.4 (5.2–5.6)
Uric acid (mg/dL)	4.65 (3.9–6)	4.55 (3.7–5.8)
Testosterone (only for men) (ng/mL)	4 (3.98–5.3)	4.5 (3.9–17)

* All values are summarized by median and interquartile range; kilograms: kg; m^2^: meters squared; Adiposity-Based Chronic Disease: ABCD; Dual X-ray absorptiometry: DEXA; BMI: body mass index.

**Table 3 ijerph-19-12406-t003:** Clinical characteristics during teleconsultations at months 3 and 6 by type of evaluation.

	First-Time Group (FG), n = 88	Follow-Up Group (FUG), n = 114
Weight (kg)		
Value at 3 months	75 (67.1–81.7)	73.8 (66–88)
Value at 6 months	72.5 (63–81)	70.5 (60–83)
BMI (kg/m^2^)		
Value at 3 months	27.9 (26.1–30.9)	28 (25.6–31.2)
Value at 6 months	26.6 (24.8–30.3)	26.35 (24.2–29.2)
Fasting blood glucose (mg/dL)		
Value at 3 months	91.7 (89–105)	90 (83.5–95)
Value at 6 months	92 (86–98.5)	89.5 (86–91)
LDL cholesterol (mg/dL)		
Value at 3 months	96 (73–127)	106 (83–134.3)
Value at 6 months	99.5 (79–116)	118 (64–135)
HDL cholesterol (mg/dL)		
Value at 3 months	53 (46–62)	49 (43.25–57.5)
Value at 6 months	45 (41–49)	51 (42.1–72)
Triglycerides (mg/dL)		
Value at 3 months	92 (88–130)	119 (93–146)
Value at 6 months	142.5 (109–164)	97 (86–145)
Creatinine (mg/dL)		
Value at 3 months	0.815 (0.7–0.9)	0.8 (0.71–0.9)
Value at 6 months	0.77 (0.7–0.9)	0.8 (0.6–0.8)
Ferritin (ng/mL)		
Value at 3 months	66 (31–133)	130 (79–249)
Value at 6 months	204 (30.9–625)	175 (12–268)
Glycosylated hemoglobin (%)		
Value at 3 months	5.4 (5.2–5.8)	5.42 (5.1–5.5)
Value at 6 months	5.4 (5.37–5.5)	5.4 (5.2–5.6)
Uric acid (mg/dL)		
Value at 3 months	3.9 (3.6–4.8)	3.62 (3.38–4.2)
Value at 6 months	4.2 (4.1–4.8)	4.3 (3.56–4.6)
Testosterone (only for men) (ng/mL)		
Value at 3 months	4.75 (4.75–4.75)	6.735 (3.07–10.4)
Value at 6 months	-	4.11 (4.11–4.11)
Percentage of body fat by DEXA evaluated at last teleconsultation (%)	41.25 (37–47)	48 (45–51)
Total fat weight by DEXA evaluated at last teleconsultation, kg	32 (26–34.3)	42.85 (36–49.7)
Android fat/gynoid fat ratio by DEXA at last teleconsultation	0.965 (0.92–1.06)	0.96 (0.91–1.01)
Physical activity at last teleconsultation (%)		
No	14	10
Yes	42	58
ND	32	46
Healthy diet at last teleconsultation (%)		
No	9	9
Yes	46	63
ND	33	42

All values are summarized by median and interquartile range; kilograms: kg; m^2^: meters squared; Adiposity-Based Chronic Disease: ABCD; Dual X-ray absorptiometry: DEXA; BMI: body mass index.

**Table 4 ijerph-19-12406-t004:** Comparison of weight (kg) at each time point analyzed by age, type, and number of teleconsultations, and ABCD stage.

	Time 0, n = 78	Time 1, n = 63	Time 2, n = 36	*p* Value
General weight, kg	84.0 (73.0–97.0)	78.0 (68.0–90.0)	75.0 (68.0–88.0)	<0.001
Age, years				
18–26	82.0 (67.0–93.0)	73.0 (62.0–82.0)	78.5 (67.5–98.5)	<0.001
27–59	84.0 (73.0–96.0)	78.5 (69.0–90.0)	73.0 (68.5–84.5)	<0.001
≥60	98.5 (84–103)	99.5 (89–102)	101 (55–104)	>0.99
Type of teleconsultation				
First time	83.0 (73.0–90.0)	78.0 (68.0–85.0)	71.5 (63.5–83.0)	<0.001
Control	89.0 (73.0–101.0)	82.5 (68.0–100.0)	82.0 (73.0–99.0)	<0.001
Number of teleconsultations				
2–3	84.0 (73.0–102.0)	79.0 (68.0–100.0)	79.0 (69.0–94.0)	<0.001
4–5	84.0 (73.0–91.0)	75.0 (71.0–87.0)	72.0 (63.0–82.0)	<0.001
≥6	84.0 (73.0–97.0)	79.0 (67.5–91.5)	81.5 (69.0–88.0)	<0.001
ABCD stage				
0	75.0 (66.0–85.0)	69.0 (62.0–76.0)	69.5 (63.0–80.0)	<0.001
1	90.0 (80.0–102.0)	84.0 (76.0–98.0)	81.0 (73.0–94.0)	<0.001
2	104.0 (101.0–122.0)	103.0 (100.0–117.0)	93.0 (68.5–112.0)	0.215

For the construction of the model, the following variables were considered: age, healthy diet habits, physical activity, attention type, and number of attentions via telemedicine. However, there were no statistically significant differences and were not considered confounding variables for the weight changes identified over time.

**Table 5 ijerph-19-12406-t005:** Weight change over time fitted with a random intercept linear regression model.

Factor	Adjusted Weight Change	*p* Value
Time		
Time 0	-	
Time 1	−4.1 (−5.1–−3.1)	<0.001
Time 2	−8.6 (−10–−7.3)	<0.001
ABCD stage		
0	-	
1	15.6 (10.1–21.1)	<0.001
2	28.2 (15.7–40.8)	<0.001

95% (CI).

## Data Availability

The datasets analyzed during the current study and that support the findings of this study are available from Fundación Valle del Lili (FVL) but restrictions apply to the availability of these data, due to internal privacy policies. Data are, however, available from the authors upon reasonable requests and with permission of Fundación Valle del Lili.

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
