# Peer review of "Experience with Obese Patients Followed via Telemedicine in a Latin American Tertiary Care Medical Center"

_ijerph, 2022, doi:10.3390/ijerph191912406_

Round 1

Reviewer 1 Report

GENERAL COMMENT

This study may be of interest to the readers of IJRPH, but it needs substantial improvement as reporting and statistical analysis are concerned. One statistical method should be ultimately chosen to analyze the data, and missing data should be clearly reported and handled.

MAJOR COMMENTS

L20 and elsewhere: it is not clear to me why the Skilling Mack-test was used to compare medians over time. Was this done because of missing data? (The Skilling Mack-test is roughly equivalent to the Friedman test in the absence of missing data.) Note that the Skilling Mack-test was proposed for missing block (and other) designed studies, but it is not a  good choice for the present study design (retrospective cohort):

https://www.ncbi.nlm.nih.gov/pmc/articles/PMC2761045/

The random intercept linear regression model used to calculated mean (95%CI) weight change is much more interesting, as it provides clinically useful estimates of effect sizes (L 167). It requires normally distributed between-time differences, but also this requirement can be relaxed. I suggest selecting the random intercept linear regression model as the model of interest, as it is too confusing to present both the non-parametric (Skilling-Mack) and parametric approaches (random intercept linear regression). The random intercept linear regression can also easily handle covariables, which I suppose is what is done in Table 5. (Please specify whether such analysis is a multivariable analysis.) With some caution, the random intercept linear regression can also handle missing data (see below).

L132 and elsewhere: please provide units of measurement.

L148 Please provide an adequate description of missing data. Here you say that 78 patients were available at baseline, 63 at time 1, and 36 at time 2, to "identify the change in weight". How should, then, one interpret the "n=88" and "n=114" in Table 3? The numbers in Table 2 give the impression that the number of patients remained the same during follow-up.

L155 Figure 1 Y-axis: peso → weight; tiempo → time (please specify the time unit in the graph). It would be very nice to see the individual trajectories over time, also because the random intercept linear regression model relies on them.

MINOR COMMENTS

L18 I would describe this as a retrospective cohort study.

L132 Health insurance. What is meant with "particular"?

L132 How can DEXA be performed during a teleconsultation? This is probably just a matter of grammar.

L132 please be consistent with the number of numbers of decimals, e.g., 142.5 (109-164) vs. 97 (86-145) should be 142 (109-164) vs. 97 (86-145) or whatever you wish.

L5 Please delete "Haga clic o pulse aquí para escribir texto"

L132 Why did you evaluate ferritin and testosterone?

L132 Please specify the package used to perform statistical analysis. Figure 1 seems to have been produced by Stata.

L237 What do you mean by saying that other variables were not assessed individually?

Author Response

analysis of the variables that were part of the model was carried out, for which a Skilling Mack test was conducted in order to evaluate the behavior of the variables and make the adjustments to build the regression model. A random intercept regression model was used because patients were measured on different occasions.

4 . In response to the comment “ L148 Provide Units of measurement”: The units of measurement in table 1 are percentages and the total of patients from each group, in table 2 the measures are the following, and it was also corrected in the manuscript.

Weight: Kilograms (Kg)
Body Mass Index: (Kg / m2)

Fasting Blood Glucose: (mg/dl)

LDL Cholesterol (mg / dl)

HDL (mg/ dl)

Ferritin ng/ mL

Glycosylated Hemoglobin %

Uric Acid mg/dl

Testosterone ng/ mL

  1. Giving response to the comment: “Please provide an adequate description of missing data. Here you say that 78 patients were available at baseline, 63 at time 1, and 36 at time 2, to "identify the change in weight". How should, then, one interprets the "n=88" and "n=114" in Table 3? The numbers in Table 2 give the impression that the number of patients remained the same during follow-up”.

To build graph 3 as well as the box plot that identifies the changes in weight, we included the patients who were adherent to the consultations and had the measures that allowed us to estimate the changes in weight, this clarification was made in the manuscript.

  1. According to the comment: “Figure 1 Y-axis: peso → weight; Tiempo → time (please specify the time unit in the graph). It would be very nice to see the individual trajectories over time, also because the random intercept linear regression model relies on them”.

The graph of the individual trajectories by times is represented in the following Graph. ( See Attach file)-

However we consider this graphic must be confusing for the readers, we corrected the graphic placed in the manuscript

  1. According to the comment “I would describe this as a retrospective cohort study”.

We agree with this comment, it has already been changed in the abstract and methods section L18-L68

  1. Giving response to “What is meant with "particular"?

Particular means: Patients who come to the institution and pay for medical services in a private way, using their own economic resources.

Subsidized: Patients whose health care expenses are covered by the state directly, according to their financial and employment situation.

Contributory: Patient who is making periodic economic contributions to entities that provide health services to cover their health care.

  1. Giving response to the comment “How can DEXA be performed during a teleconsultation? This is probably just a matter of grammar”.

We agree with the reviewer, it is a grammatical problem, we made a correction in the manuscript, and we clarified in the methods section that the Dual-energy X-ray absorptiometry (DXA) was ordered by the endocrinologist, as well as the other paraclinical studies that the patient’s paraclinical studies that the patient had to undergo to attend the next consultation.

* Percentage of body fat by DEXA evaluated at last teleconsultation (%)

*Total fat weight by DEXA evaluated at last teleconsultation, kg

  1. Please be consistent with the number of numbers of decimals, e.g., 142.5 (109-164) vs. 97 (86-145) should be 142 (109-164) vs. 97 (86-145) or whatever you wish.

We agree with this comment, we already fixed this in the manuscript.

  1. In response to the following comment “L132 Why did you evaluate ferritin and testosterone?”

The explanations are listed below, and we also decided to include them in the overview of the processes associated with care.

Ferritin was one of the paraclinical tests routinely evaluated in the consultation, because its measured estimates iron stores that are often increasing due the metabolic dysfunction, and its related to the onset of Non-Alcoholic Fatty Liver Disease predisposing to other associated conditions such as malignancy, liver dysfunction, and cirrhosis (Trasolini et al., 2022).

Testosterone was routinely evaluated only in male patients because it is a key hormone in metabolic diseases such as obesity, where decreased levels are associated with increased corporal mass, especially central, and an energy imbalance with metabolic effects such as decreased insulin sensitivity and dyslipidemia (Kelly & Jones, 2015).

  1. L132 Please specify the package used to perform statistical analysis. Figure 1 seems to have been STATA?

The data analysis was performed using the software R (R Core Team, v. 1.4.1717) with the nlme library, We performed a Stata data analysis for descriptive statistics.

References:

Kelly, D. M., & Jones, T. H. (2015). Testosterone and obesity. Obesity Reviews, 16(7), 581–606. https://doi.org/10.1111/obr.12282

Trasolini, R., Cox, B., Galts, C., Yoshida, E. M., & Marquez, V. (2022). Elevated serum ferritin in non-alcoholic fatty liver disease is not predictive of fibrosis. Canadian Liver Journal, 5(2), 152–159. https://doi.org/10.3138/canlivj-2021-0002

Reviewer 2 Report

Summary of article:

Dr. López et al. investigated the effectiveness of the telemedicine program for people with obesity. The authors conducted the retrospective cohort study, which included 202 patients. As a result, the authors found that the telemedicine program showed good adherence and weight change at the follow-up. They also found that the effect size of the telemedicine program was -4.1kg at three months and -8.6kg at six months. They concluded that the telemedicine program could be a helpful tool to support the continuity of care and shows satisfactory results in the management of obese patients.   

Comments (Invitation on Aug 9, 2022, and comment submission on Aug 16, 2022)

This study addressed an interesting and hot topic about the telemedicine program for people with obesity. I would like to congratulate all authors’ efforts in this study. Please consider addressing some concerns as shown below.

Here are my comments and suggestions about this manuscript.

Major points:

[1] “Method”:

Please clarify whether the authors conducted multiple testing corrections. Specifically, Table 4 shows the results of explanatory statistical comparison based on the stratification. This statistical comparison needs multiple testing corrections.

[2] “Method”:

Please describe the details of mixed model linear regression. What factors did the authors incorporate into fixed effect and random factor?

[3] “Method”:

Please clarify the definition of variables “physical activity at first teleconsultation” and “healthy diet at first teleconsultation.” Are there any criteria to categorize into Yes/No in each question?

[4] “Discussion”

Please discuss why the weight value was higher in Time 2 than in Time 1 in the group of age 27-59, >6 times of teleconsultations, and ABCD stage 0.

The authors might want to consider the effect of rebounding or getting worse in compliance with weight-loss activities.

[5] “Discussion”

Please add other limitations of this study. For example, one of the limitations of this study is the lack of the control group as in-person consultation or non-intervention. Please note that this study is based on a self-controlled study design.

Minor points:

[6] “Table 2”

Readers cannot understand why the row of stage 2 in the First-time group is blank and the summation in the First-time group is not 88 (43+44=87).

[7] “Table 4”

Please replace the p-value “1.00” with “>0.999”. This is because the probability cannot be 100% in statistics.

[8] “Table 5”:

Please clarify what the values in the parentheses stand for. I guess those mean the 95% confidence interval of effect size, but the authors should describe those in a footnote of the table.

Author Response

Dear Reviewer

International Journal Of Environmental Research and Public Health

we agree with the reviewer, it was already included in the footer of the table:

  1. Giving response to your comment:” Please clarify whether the authors conducted multiple testing corrections. Specifically, Table 4 shows the results of explanatory statistical comparison based on the stratification. This statistical comparison needs multiple testing corrections”.

The differences in the responses can be seen in the graph [Attached File]; however, there is no consensus in the reviewed literature on applying a posthoc analysis for the Skillings-Mack test, nor is there a function or library implemented in the statistical program to perform this process.

2. Regarding the comment “Please describe the details of mixed model linear regression. What factors did the authors incorporate into fixed effect and random factor?”

The random intercept was used to control for inter-patient variability. Patients were measured at different points in time, fixed factors are variables that do not change over time (Age, type of consultation, ABCD status).

3.Regarding the comment: “Please clarify the definition of variables “physical activity at first teleconsultation” and “healthy diet at first teleconsultation.” Are there any criteria to categorize into Yes/No in each question?”

Since it was a retrospective study, the data were taken from the medical records of the consultation where patients were asked questions about eating habits and physical activity in their daily lives, the YES/NO categorization was subjectively determined by the clinician, and an instrument was not used.

This will be explained within the overview of the consultation processes within the manuscript.

4. In response to the comment: “Please discuss why the weight value was higher in Time 2 than in Time 1 in the group of age 27-59, >6 times of teleconsultations, and ABCD stage 0. The authors might want to consider the effect of rebounding or getting worse in compliance with weight-loss activities”.

We agree to the reviewer, although for time 1 the weights decreased with respect to time 0 in this group, the increase in weight observed at time 2 can be considered part of the rebound effect, one of the challenges of this type of care is that the patients do not return to baseline levels. We will address this in the discussion.

5. About the comment: Please add other limitations of this study. For example, one of the limitations of this study is the lack of the control group as in-person consultation or non-intervention. Please note that this study is based on a self-controlled study design.

 We agree with the reviewer, we include this within our limitations

6. According to the comment related to :"[Table 2] Readers cannot understand why the row of stage 2 in the First-time group is blank and the summation in the First-time group is not 88 (43+44=87)".

We agree with this comment, reviewing our data, we found a stage 2 patient that had not been written in the table of the manuscript, however, it was already corrected in the manuscript.

7.  According to the comment related to:"[ “Table 4”]Please replace the p-value “1.00” with “>0.999”. This is because the probability cannot be 100% in statistics.

We agree with the reviewer, we already corrected it in the table placed in the manuscript.

8. Please clarify what the values in the parentheses stand for. I guess those mean the 95% confidence interval of effect size, but the authors should describe those in a footnote of the table.

We agree with the reviewer, it was already included in the footnote of Table 5 as it was suggested.

Kind Regards

Maria Fernanda Escobar MD Msc

Correspondence Author

Round 2

Reviewer 1 Report

The overall presentation of the manuscript is improved. It puzzles the statistician in me—however—that the non-parametric Skillings-Mack (SM) test is still being used together with the parametric random effect linear regression model (RE-LRM). 

In detail: 

1) The much more useful RE-LRM can easily estimate the effect sizes of interest, i.e., the (time 1-time 0) and (time 2-time 1)—or other—contrasts. Statistical significance should be obtained *from such contrasts*, not from the SM test. Or, if you choose to stay with the SM test, you should renounce to a meaningful metric of effect size. You will, in fact, obtain only a p-value from the Skillings-Mack test, besides a number of (sums of) ranks without direct clinical interpretation. 

2) The Skillings-Mack test can be used to handle missing data provided that they are missing at random (MAR), but does not allow correction for covariables potentially explaining such missingness. Also in this regard, the RE-LRM is superior to the SM test. Whatever the case, you should say something about how likely your data is to be MAR. (There is no formal test for that.) 

3) Note that figure 1 gives just the *inter-individual* variability in weight at the given time point. Both RE-LRM and SM do consider of course *intra-individaul* variability together with inter-individual variability. 

4) I really do not understand how you did the following: "A preliminary analysis of the variables that were part of the model was carried out, for which a Skilling Mack test was conducted *in order to evaluate the behavior of the variables and make the adjustments to build the regression model*. A random intercept regression model was used because patients were measured on different occasions".

(Note that Skilling is Skillings) 

I have never heard of a variable selection procedure for a parametric model performed by using a non-parametric model. And I think there is a good non-philosophical reason for that: you will get no effect size besides a scarcely useful p-value (especially without a pre-specified null hypothesis, as in the present case).

5) I strongly suggest basing your inferences on the RE-LRM. The choice is yours. However, it is clear that the non-parametric and parametric approaches should not be presented together. Or, *at the very least*, avoid declaring as statistically significant differences obtained from the RE-LRM but with a formal evaluation performed by the SM test.  This is not acceptable.

Author Response

Santiago De Cali, September 21/2022

Dear Editors and Reviewers

International Journal of Environmental Health and Public Health 

  • Giving response to the comment:

“The much more useful RE-LRM can easily estimate the effect sizes of interest, i.e., the (time 1-time 0) and (time 2-time 1)—or other—contrasts. Statistical significance should be obtained *from such contrasts*, not from the SM test. Or, if you choose to stay with the SM test, you should renounce to a meaningful metric of effect size. You will, in fact, obtain only a p-value from the Skillings-Mack test, besides a number of (sums of) ranks without direct clinical interpretation”

We agree with the comment, in this review we gave a new approach to the statistical analysis. Initially, a descriptive analysis was performed to characterize the population of patients included in the study. Subsequently, the Skillings Mack test was performed to perform an analysis by subgroups of interest, which compares the observed differences in patient weights at the three points in time.

To estimate the average change of weight over time, we performed a mixed linear regression model with random intercept used to control inter-patient weight variability. For this purpose, we relied on a propositional variable selection model, which has been referenced in the manuscript.

  • Regarding the following comment:

“The Skillings-Mack test can be used to handle missing data provided that they are missing at random (MAR) but does not allow correction for covariables potentially explaining such missingness. Also in this regard, the RE-LRM is superior to the SM test. Whatever the case, you should say something about how likely your data is to be MAR. (There is no formal test for that.)”

We agree the RE-LRM will always be superior, in this new version it is shown independent of the Skillings Mack test, we decided to leave table 4 because we find the analysis by subgroups interesting and informative.

3) Regarding the following comment:

“I really do not understand how you did the following: "A preliminary analysis of the variables that were part of the model was carried out, for which a Skilling Mack test was conducted *in order to evaluate the behavior of the variables and make the adjustments to build the regression model*. A random intercept regression model was used because patients were measured on different occasions".

(Note that Skilling is Skillings) 

I have never heard of a variable selection procedure for a parametric model performed by using a non-parametric model. And I think there is a good non-philosophical reason for that: you will get no effect size besides a scarcely useful p-value (especially without a pre-specified null hypothesis, as in the present case).

This part was corrected in the manuscript, we agree that it was not adequately addressed and described at the beginning, refer to page 3 in the methods section where this was corrected.

the method of variable selection used can be explored in the following article:

https://scfbm.biomedcentral.com/articles/10.1186/1751-0473-3-17

5) Giving response to the comment “ I strongly suggest basing your inferences on the RE-LRM. The choice is yours. However, it is clear that the non-parametric and parametric approaches should not be presented together. Or, *at the very least*, avoid declaring as statistically significant differences obtained from the RE-LRM but with a formal evaluation performed by the SM test.  This is not aceptable”.

We agree with the reviewer and in the response to point 1 we summarize the new approach to the use of the model.

Your comments and observations have been of great help in reorganizing our ideas and correcting our work with great effort.

Kind Regards

Maria Fernanda Escobar MD MsC
